# Structural Basis for the Regiospecificity of a Lipase from *Streptomyces* sp. W007

**DOI:** 10.3390/ijms23105822

**Published:** 2022-05-22

**Authors:** Zexin Zhao, Siyue Chen, Long Xu, Jun Cai, Jia Wang, Yonghua Wang

**Affiliations:** 1Key Laboratory of Fermentation Engineering (Ministry of Education), Hubei Key Laboratory of Industrial Microbiology, Hubei University of Technology, Wuhan 430068, China; zx_zhao@hbut.edu.cn (Z.Z.); caijun@mail.hbut.edu.cn (J.C.); 2School of Food Science and Engineering, South China University of Technology, Guangzhou 510640, China; chensiyuecsy@hotmail.com; 3College of Food Science and Technology, Henan Agricultural University, Zhengzhou 450002, China; xulong19891026@163.com; 4College of Life Science, Guangzhou University, Guangzhou 510006, China

**Keywords:** lipase, structure-function relationship, substrate-binding pocket, protein engineering, mutagenesis study

## Abstract

The efficiency and accuracy of the synthesis of structural lipids are closely related to the regiospecificity of lipases. Understanding the structural mechanism of their regiospecificity contributes to the regiospecific redesign of lipases for meeting the technological innovation needs. Here, we used a thermostable lipase from *Streptomyces* sp. W007 (MAS1), which has been recently reported to show great potential in industry, to gain an insight into the structural basis of its regiospecificity by molecular modelling and mutagenesis experiments. The results indicated that increasing the steric hindrance of the site for binding a non-reactive carbonyl group of TAGs could transform the non-specific MAS1 to a α-specific lipase, such as the mutants G40E, G40F, G40Q, G40R, G40W, G40Y, N45Y, H108W and T237Y (PSI > 80). In addition, altering the local polarity of the site as well as the conformational stability of its composing residues could also impact the regiospecificity. Our present study could not only aid the rational design of the regiospecificity of lipases, but open avenues of exploration for further industrial applications of lipases.

## 1. Introduction

Triacylglycerol (TAG) lipases (EC 3.1.1.3) are versatile biocatalysts, and the high regiospecificity is one of the most attractive properties of the enzymes [1,2,3]. Although the α-specific lipases (preferring the *sn*-1 and/or *sn*-3 positions of glycerol backbone of TAGs) are wildly used in the production of structured lipids that refer to a new type of functional acylglycerols with special molecular structures [4], the regiospecific mechanism of lipases still remains ambiguous. The revelation of the structural basis for lipase regiospecificity would not only enrich the enzymology theory, but also pave a way for designing the tailor-made lipases by artificial intelligence in the future.

According to the position specificity index (PSI) definite by Miwa et al., lipases are classified into two groups: α-specific (PSI ≥ 80, such as lipases from Porcine pancreatic, *Aspergillus niger*, *Penicillium camemberti*, *Rhizomucor miehe* and *Saccharomycopsis lipolytica*) and non-specific (|PSI| < 80, such as lipases from *Candida cylindracea*, *Geotrichum candidum* and *Candida antarctica*), whereas β-specific lipases (PSI ≤ −80) have not been found yet [5,6]. The previous works have well described the structural basis of the stereospecificity toward TAG (analogs) and other compounds in lipases from *Burkholderia cepacia, Pseudomonas aeruginosa*, *Thermomyces lanuginosus*, *Candida rugosa*, *Rhizopus oryzae*, *Rhizopus delemar*, etc., and their stereospecific modification have been achieved by protein engineering [7,8]. However, their ability to discriminate between outer α (*sn*-1/3) and inner β (*sn*-2) positions of the TAG backbone has not been well recognized.

Recently, our laboratory characterized a non-specific and thermostable lipase from marine *Streptomyces* sp. W007, MAS1, which exhibits great potential in the production of *n*-3 PUFA-enriched TAGs and biodiesel [9,10,11]. Meanwhile, we resolved the crystal structures of MAS1 and constructed its α-specific mutant H108W [12]. Thus, MAS1 is a preferable template for further revealing the molecular basis of lipase regiospecificity.

Here, we constructed the binding conformations of tricaprylin (TC) in different reaction modes to MAS1 and the mutant H108W by molecular docking. These conformations were comprehensively compared to explain the sketchy molecular basis of the regiopreference conversion between the two enzymes. To further understand the structural basis for regiospecificity of MAS1, the detailed binding geometry of MAS1-TC complex in *sn*-2 reaction mode and the key residues G37, N45 and H108 for binding a non-reactive carbonyl group of the substrate were determined by the molecular dynamic simulation. A serial of mutants at the residues G40, N45, H108 and T237, whose side chains formed the binding site of non-reactive carbonyl group, were constructed and characterized to investigate the effect of structural modification in this region on the regiopreference of MAS1. In addition, in silico simulations were performed to shed light on the molecular mechanism varying the regiospecificity of mutants. Our studies would pave a way for designing the regiospecificity-oriented lipases by rational protein engineering.

## 2. Results

### 2.1. Catalytic Pocket of MAS1

Structural superimposition of the open conformations of MAS1 and PAL (PDB ID: 1EX9) as well as BCL (PDB ID: 5LIP) showed that the catalytic cores of these enzymes was similar (Figure 1A). The complex structures of PAL and BCL with an analogue of TC (presenting a tuning fork shape) suggested that each catalytic pocket could be divided into three parts to accommodate the three acyl chains of TAG [13,14]. Based on the structural similarity, the catalytic groove of MAS1 was also divided into three parts termed pocket A, B and C (Figure 1B). The reactive acyl chain of TAG could fit snugly into the valley-like pocket A constituted by residues T38, L143, L144, Q170 and V202, while pocket B (comprised of residues G37, G40, D44, N45, F74, H108, V233 and T237) and C (comprised of residues F39, V202 and V233) could be used for binding the other two non-reactive acyl chains of TAG.

### 2.2. Comparison of Different Reaction Conformations of TC

Due to the prochirality of TAG, three isomeric DAGs could be produced in its deacylation processes. Therefore, in theory, there are three different reaction modes of TAG during its lipase-catalyzed deacylation. To gain some insights into the molecular basis of the regiospecificity of MAS1, the reaction conformations of TC in *sn*-1, *sn*-2 as well as *sn*-3 reaction modes were manually built using the covalent binding tricaprylin (TC) analogue of PAL and BCL mentioned before as the templates (see Appendix A for the atom numbering of TC in different conformations). Close inspection of the binding geometries between the substrate and MAS1 in all modes showed a similar position of the reactive octyl chains in pocket A (Figure 1C, the catalytic triad was formed by the hydrogen bond linked S109, H232 and D200). Moreover, in all modes, the distances between the nitrogen atoms in the main chain of T38 and Q110 (oxyanion hole) and the carbonyl oxygen of reactive octyl chains were ~3.0 Å and ~3.5 Å, respectively, and the distances between Oγ of the catalytic serine and their carbonyl carbon were ~3.0 Å. However, the non-reactive octyl chains accommodated by pocket B presented two types of configurations in three different modes. As shown in Figure 1D, the TC C1 in the *sn*-2 reaction mode was more than 2 Å below the TC C2 in the *sn*-1/3 reaction modes, indicating the glycerol backbone of TC in the former mode being closer to the bottom of the pocket B than that in the latter.

### 2.3. Binding Geometry of TC in the sn-2 Reaction Mode

To investigate the detailed mechanism of the β-activity of the non-specific MAS1, which was the precondition of its β-preference, a 50 ns MD simulation of MAS1-TC complex with the *sn*-2 reaction conformation was performed. The backbone RMSD (root-mean-square deviation) analysis showed that the catalytic core domain (excluding the residues K148-T164 located in the lid domain) was stable during the simulation, while the overall conformation achieved balance after 30 ns due to the displacement of the loop region in the lid (Appendix A). Although the substrate TC was relatively flexible (Appendix A), the position of the reactive acyl (*sn*-2 moiety) and the glycerol backbone of the substrate underwent negligible changes (Appendix A). Thus, the fluctuation of TC was mostly derived from the two less restrained non-reactive chains.

Hydrogen-bond network between enzyme and substrate plays an important role in substrate binding and reaction. During the simulation, the amide group of N45 formed H-bonds with the NH group of N41 and the O12 of TC at the end frame, respectively (Figure 2A). Additionally, an H-bond connected G37 and TC was also found in the last frames (Figure 2A). As shown in Figure 2B, five H-bonds between MAS1 and TC existed stably including the two mentioned above (G37 NH…TC O12 and N45 NδH…TC O12) and the other two involved in the oxyanion hole (T38 NH…TC O22 and Q110 NH…TC O22). An almost consistent H-bond distance between Ser109 and TC (Ser109 OγH…TC O21) was found during the whole simulation process (Figure 2B). The hydroxyl of Ser109 simultaneously kept two H-bonds with H232 and TC for facilitating proton transfer from Oγ of the serine to TC O21 (Figure 2A, and see Appendix A for the detailed reaction mechanism). Thus, the appropriate geometry of the catalytic pocket and the H-bonds from the surrounding amino acids interacted with the substrate constituted of the molecular basis for the high β-activity of MAS1.

### 2.4. Regiospecificity of Mutants

To further investigate the roles of the geometry and the microenvironmental polarity of this region in the regiospecificity of MAS1, the substitutions of the residues composed the region (including G40, N45, H108 and T237) were performed. All G40 mutants including G40E, G40F, G40Q, G40R, G40W and G40Y were transformed to α-specific lipases (the PSIs of all these mutants changed from −54 of wild type to over 90, see Figure 3). Compared with wild-type enzyme, the mutants N45A, N45E, H108A, H108E, T237F and T237Q lost the β-preference, while the mutants N45D, N45R, H108Q, T237A and T237E remained its regiospecificity (Figure 3). Interestingly, besides H108W and G40 mutants, the mutants N45Y, H108F, H108Y, T237R and T237Y were also converted into α-preference (Figure 3), but the mutant H108W showed a highest PSI (>99). It should be pointed out that the regiospecificity of MAS1 and its mutant H108W expressed in *Pichia pastoris* were similar to that of them expressed in *E. coli* [12], indicating that the regiospecificity of lipases may be few affected by the expression systems.

### 2.5. Substrate Binding Stability

To explore the mechanism governing the different regiospecificity of mutants, the enzymes with non-specificity including N45A, N45D, N45E, N45R, H108A, H108E, H108Q, H108Y, T237A, T237E, T237F and T237Q (PSI < 40) were chosen to perform the enzyme-substrate complex (*sn*-2 reaction conformation) MD simulations. Except H108Y, all these mutants could bind the substrate stably (the detail substrate binding conformations and trajectory analysis see Appendix A). The interaction energy analysis between substrate and enzymes showed that the binding energies of N45D, N45R and T237A were close to that of wild type, which was due to the mutual cancellation of changes in electrostatic and van der Waals interaction energy caused by mutation (Figure 4). The decreases in electrostatic energy of T237E and T237Q were lower than the increases in their van der Waals item, resulting in the reductions of their binding energy by 7.55 and 14.77 kJ/mol (Figure 4). On the contrary, the binding energy of H108Q increased by 13.25 kJ/mol (Figure 4). With regard to the mutants N45A, N45E, H108A, H108E and T237F, their binding energy were decreased because of the reduction of the two interaction energy items (Figure 4).

### 2.6. Effect of Chain Length on the Regiospecificity

It was documented that lipases exhibited different regiospecificity for different natural oils as substrates [15]. MAS1 and its mutants G40E, N45R, H108W, T237E and T237Q displayed various regiospecificity during catalyzing the hydrolysis of TC (Figure 3). To investigate the effects of chain lengths of fatty acids on lipase regiospecificity, we measured their PSIs using trilaurin and triolein as substrates and compared them with the PSIs determined by hydrolyzing TC. As shown in Figure 5, the regiospecificity of these lipases was barely affected by chain lengths of fatty acids variations. Thus, it was reasonable to use TC as a patterned substrate in the MD simulation studies. Since different fatty acids were distributed at various sites of the glycerol backbone in natural oils, the previous results appeared to be caused by the combined effects of fatty acid selectivity and regiospecificity of lipases.

## 3. Discussion

In nature, lipases present diverse regiopreference toward the glycerol backbone of TAG. Some lipases with α-specificity can be used for producing lipids with specific molecular structure, others without apparent regiopreference are potential biocatalysts for the production of high-value TAGs via esterification/transesterification and of biodiesel via alcoholysis of TAGs [4,10,11,16,17]. Though the phenomenon has long been observed, the structural basis governing their different regiospecificity is still not recognized well. In our previous work, through multiple sequence and structural alignment, we determined that the bulk of amino acid located at the second site of conserved pentapeptide of bacterial lipase I.7 subfamily played an important role in their regiospecificity, and the substitution of the H108 (the second residue of conserved pentapeptide) of MAS1 that belonged to this lipase subfamily with tryptophan conversed its non-specificity to α-specificity [12].

### 3.1. Role of H108

In this paper, three different reaction modes of TC with MAS1 were constructed firstly, and the binding geometries of TC in *sn*-1/3 and *sn*-2 reaction modes presented significant difference (Figure 1C,D). It might explain how the regiospecificity change of MAS1 after mutating H108 to tryptophan. Although TAG is a type of flexible compound with many rotatable bonds, when the *sn*-2 acyl chain of TAG bound to the pocket A of MAS1, the position of glycerol backbone (including C1 and C3) was relatively fixed due to no enough space in the narrow catalytic cavity to adjust its conformation (Figure 1D). When H108 was substituted with a bulkier tryptophan, the minimum distance between the residue 108 and TC C1 in the *sn*-2 reaction mode decreased from 3.5 Å to 2.8 Å, which formed a strong steric hindrance to the glycerol backbone, resulting in the unfavorable binding of TAG to the mutant H108W. The distance between W108 and TC C2, in comparison, was ~4.0 Å in the *sn*-1/3 reaction modes (Figure 1D), which hardly impede the entrance of substrate to binding site. Consequently, the mutant H108W displayed a narrower regioselectivity.

Besides preserving the binding space, the residue H108 may play other roles in the β-preference of MAS1. Similar to the close contact between Oγ of catalytic S109 and the partially positively charged TC C21 (3.19 ± 0.17 Å, for the subsequent nucleophilic attack), the Nδ1 of H108 also kept a distance of 3.24 ± 0.22 Å from TC C11 (Figure 2C). It was documented that both serine and histidine acted as nucleophilic reagents in enzymatic reactions [18,19]. Moreover, both N45 and G37 formed an H-bond with TC O12 (Figure 2A), which seemingly constituted another oxyanion hole. However, the alanine substitution of the catalytic serine (S97) or histidine (H232) resulted in the complete inactivation of MAS1, implying residue H108 did not act the nucleophilic reagent in here (data not shown). To further explore the functions of H108, a series of mutants including H108A, H108E, H108F, H108Q, H108W and H108Y were expressed and characterized. The PSI of the mutant H108A changed from −54 to −4 (Figure 3). The MD trajectory of TC showed that the track of TC O12 in the wild-type model was relatively convergent as compared to that in H108A (Appendix A). It was found that the H-bond between G37 and TC O12 became unstable in the H108A, and the H-bond between N45 and TC O12 was not no longer easy to form (Appendix A). When H108 was substituted with alanine, the electrostatic and van der Waals interaction energy between the enzyme and TC were sharply decreased by 18.21 and 17.45 kJ/mol, respectively (Figure 4). These observations indicated that H108 played a key role in the orientation and binding of TAG when the substrate was bound in a *sn*-2 reaction mode.

Besides H108A, the mutant H108E also lost the β-preference (Figure 3). It was found that the side chain of N45 formed an H-bond with N41, E108 and TC, and G37 kept an H-bond distance with TC in the mutant (Appendix A). These interactions were supposed to stabilize the binding conformation. However, a remarkable conformation fluctuation was observed herein because of the repulsion caused by positional proximity between the negatively charged carboxyl group of E108 and TC O12 (3.38 ± 0.38 Å, see Appendix A). While in H108Q, the steady conformation of the *sn*-1 moiety of TC was due to the formation of two strong and stable H-bonds by Q108 with TC O12 and H232 (the bond lengths were 1.90 ± 0.15 and 2.11 ± 0.18 Å, respectively. See Appendix A). Compared with wild type, the electrostatic energy of both H108E and H108Q decreased, while the total interaction energy of H108Q increased due to the increment of van der Waals energy (Figure 4). Thus, H108Q retained the regiopreference of wild type MAS1, in comparison, H108E changed to a non-specific lipase (Figure 3). It was worth noting that the increased interaction energy between H108Q and TC did not further enhance the β-preference of MAS1, which was presumably attributed to the longer and less stable distance of nucleophilic attack between Oγ of Ser109 and TC C21 (3.19 ± 0.17 in wild type and 3.35 ± 0.38 Å in H108Q, see Figure 2C and Appendix A).

As shown in Figure 6A, the mutations of H108 to phenylalanine, tryptophan and tyrosine all introduced a large steric hindrance in pocket B of MAS1, which impeded the substrate binding in a *sn*-2 reaction mode. All the three mutants exhibited α-preference (Figure 3), however, the PSI of the two mutants H108F (78) and H108W (99) were much higher than that of H108Y (37). Structural analysis demonstrated that the substrate could be stabilized by the H-bond between TC O11 and the hydroxyl group of Y108 (Figure 6A). Nevertheless, the H-bond was easily broken by the conformation change of Y108 (H232 and G236 competed with TC O11 for the hydroxyl group of Y108, see Appendix A). Thus, the β-activity of MAS1 could be weakened by increasing steric hindrance from the bottom of pocket B, whereas it could also be retained by improving the interaction between the enzyme and substrate.

### 3.2. Role of N45

In addition to the residue H108, the side chain of the residue N45 also participated in the binding of TC in a *sn*-2 reaction mode by forming an H-bond with O12 of the substrate (Figure 2A). To gain a comprehensive understanding of the functions of the residue N45, we characterized the regiospecificity of the mutants N45A, N45D, N45E, N45R and N45Y of MAS1. Substituting alanine for N45 led to the disappearance of the interaction and eliminated the β-preference of MAS1 (Figure 3). The MD simulation analysis showed that N45A maintained the reaction conformation of the substrate albeit a lack of H-bond interactions (Appendix A). In the view of energy, both electrostatic and van der Waals interaction between the substrate and enzyme decreased after the alanine substitution (Figure 4). Thus, the regiopreference changes of the mutants might be caused by weakened binding stability.

The other two mutants, N45D and N45E, also lost H-bonds as a result of the disappeared H-bond donors. The trajectory analysis of MD simulation showed a strong H-bond between N41 and D45 in mutant N45D (1.84 ± 0.14 Å), which effectively stabilized the surface geometry of pocket B of MAS1 (Appendix A). In contrast with N45D, the introduced residue (E45) in mutant N45E did not form any H-bond with other residues, thereby becoming more flexible (Appendix A). The flexibility of the residue not only impaired the stability of substrate binding conformation (Appendix A), but also decreased its binding energy (25.85 kJ/mol, much lower than −1.11 kJ/mol of N45D, see Figure 4). As a result, the mutants N45D and N45E presented different regiopreference.

With regard to the substitution of N45 with arginine, the PSI of the mutant N45R slightly shifted to −57 (Figure 3). The MD simulation of N4R showed that a strong H-bond network consisting of R45, G236 and TC was generated after 30 ns simulation (Appendix A). However, the increment of electrostatic interaction energy offset the reduction of van der Waals interaction energy (Figure 4). Thus, the regiospecificity of mutant N45R was similar to that of wild type. As predicted in our previous study [12], the mutant N45Y showed α-specificity (Figure 3). The introduced Y45 connected with H108 by an H-bond, which not only stabilized the local conformation but impeded the entrance of TAGs by large steric hindrance toward the glycerol backbone (Figure 6B).

### 3.3. Effect of Substitution of G40 and T237

To examine the influence of the pocket geometry on the regiospecificity of MAS1, a series of mutants regarding G40 and T237, which are located on the wall of pocket B and do not contact with the polar moiety of TAG (Figure 1B), were constructed and characterized. All G40 mutants including G40E, G40F, G40Q, G40R, G40W and G40Y were transformed to α-specific lipases (Figure 3). Structural superimposition of MAS1 and its mutants showed that the bulky side chains of F40, R40, W40 and Y40 blocked the access of substrate (*sn*-2 reaction conformation) to the catalytic site, while the carbonyl oxygens of E40 and Q40 in the according mutants occupied the binding site of TC O12 (Figure 6C,D). Thus, all these mutants could not stabilize the *sn*-2 reaction conformation of TAG substrate.

The mutant T237A retained the regiopreference of MAS1 basically (Figure 3). The MD simulation demonstrated that the substitution of T237 with alanine only resulted in slight changes of the binding geometry and energy between mutants and TC (Appendix A and Figure 4). Concerning the mutant T237F, the bulky side chain of the introduced phenylalanine compelled the *sn*-1 moiety of TC to move upwards (Figure 6E and Appendix A). The dynamic trajectory analysis revealed that the displacement of the substrate prevented the H-bond formation between TC O12 and N45 and destabilized the H-bond between TC O12 and G37 (Appendix A). The weakened interaction resulted in the elimination of the β-preference of MAS1 (Figure 3). Furthermore, bulkier side chains were introduced in the mutants T237R and T237Y (Figure 6F), leading to a substantial increase in the α-preference of MAS1 (Figure 3).

Although there were only subtle differences in the primary structure of the mutants T237E and T237Q, they showed different regiopreference (Figure 3). The analysis of simulation trajectory indicated that the residue N45 connected with N41 and E237 by H-bonds in T237E, while residues N45 and Q237 formed an H-bond with N41 and G236 in T237Q (Appendix A). Structural superimposition of the wild type with T237E and T237Q showed that N45 oriented from downwards to upwards, and the H-bond between TC O12 and N45 was not observed in both mutants (Appendix A). These changes weakened the electrostatic interaction between the mutants and substrate (Figure 4). Compared with the only H-bond formed between N41 and N45 in the pocket of T237Q, the H-bond framework formed by N41, N45 and E237 in T237E facilitated the stabilization of the substrate-binding conformation (the van der Waals interaction of T237E was stronger than that of T237Q, see Figure 4). Due to fewer connections between the residues comprising the pocket, the mutant T237Q lost more β-preference than the mutant T237E (Figure 3).

In conclusion, it was demonstrated that TAG in *sn*-2 reaction mode was more sensitive to steric hindrance from the region of MAS1 adjacent to the “nucleophilic elbow”, which is for binding a non-reactive carbonyl group of TAG. Diminishing the space of this region could enhance the α-specificity of MAS1. Additionally, on the basis of sufficient space, the strong interactions between the non-reactive carbonyl group and its surrounding residues, and the stable conformation of these residues contributed to the β-preference of MAS1. The interpretated structural basis for the regiospecificity of MAS1 in the current study would pave a way to design the regiospecificity of lipases by rational protein engineering.

## 4. Materials and Methods

### 4.1. Strains, Plasmids and Materials

*Escherichia coli* (*E. coli*) DH5α (Invitrogen, Shanghai, China) and plasmid pET22b (Novagen, Madison, USA) were used as cloning host and vector, respectively. *E. coli* Rosetta-gami B (Weidi, Shanghai, China) was used for protein expression. PrimerSTAR Max DNA polymerase and the restriction endonucleases *EcoR*I, *Xho*I and *Dpn*I were purchased from Takara (Dalian, China). Isopropyl β-_D_-1-thiogalactopyranoside (IPTG), BSA standard protein and antibiotic ampicillin were purchased from Sangon (Shanghai, China). Triolein and tricaprylin were purchased from Sigma-Aldrich (Shanghai, China), and trilaurin was purchased from TCI (Shanghai, China). Other reagents were of analytical grade and provided by a local supplier.

### 4.2. Construction of E. coli Expression Host and Protein Purification

The mas1 gene with terminus that are homologous to the multiple cloning sites region of pET22b vector was amplified by PCR using the pGAPZαA-mas1 plasmid constructed in our previous work as the template [20] (See Appendix A for the vector exchange primers). The PCR product and the expression vector pET22b were digested by restriction enzymes *EcoR*I and *Xho*I. The digested products were purified by PCR product purification kit (Sangon, Shanghai, China) and eluted by sterile H_2_O. The recombinant pET22b-mas1 plasmid was obtained by seamless assembly cloning kit (Clone Smarter, Houston, USA). The recombinant plasmid was amplified in vivo by transforming *E. coli* strain DH5α, and the plasmid was further transformed into *E. coli* strain Rosetta-gami B for expression. Transformants were grown on an LB medium plate with ampicillin (100 μg/mL). Single colonies were randomly selected to inoculate in LB medium and cultured at 37 °C with thorough agitation (200 rpm). When the OD_600_ of broth reached about 0.6, IPTG was added at a final concentration of 0.2 mM to induce the production of recombinant proteins at 20 °C for 22 h.

The cells were then collected by centrifugation at 4 °C (10,000× *g*, 15 min) and resuspended with the low imidazole concentration buffer (40 mM imidazole, 0.5 M NaCl, 20 mM NaH_2_PO_4_-Na_2_HPO_4_ pH 7.4). The target protein was released from the cells by sonication, and the supernatant after centrifugation (10,000× *g*, 15 min at 4 °C) was collected for further purification. The target protein was purified via a HisTrap HP column (GE Healthcare, Sweden). The purity of the target protein was analyzed by SDS-PAGE and the protein concentration was measured by the Bradford protein assay kit (Sangon, Shanghai, China) using BSA as a standard.

### 4.3. Mutants Constructing

MAS1 mutants including G40E, G40F, G40Q, G40R, G40W, G40Y, N45A, N45D, N45E, N45F, N45R, N45Y, H108A, H108E, H108F, H108Q, H108R, H108W, H108Y, S109A, H232A, T237A, T237E, T237F, T237Q, T237R and T237Y were constructed using the QuikChange^TM^ site-directed mutagenesis kit (Stratagene, La Jolla, CA, USA) protocol with primers listed in Appendix A. PCR products were treated with *Dpn*I followed by transformation to *E. coli* strain DH5α. The clones were grown on a LB medium plate with ampicillin (100 μg/mL). All mutant genes were confirmed by sequencing.

### 4.4. Hydrolytic Activity Assay

The determination of the hydrolytic activity of MAS1 and mutants was performed using the olive oil emulsion method [21]. One unit was defined as the amount of enzyme required to release 1 μmol of titratable fatty acid per minute under the assay conditions. 4 g of olive oil emulsion, 5 g of 100 mM phosphate buffer (pH 7.0) and 30 μL enzyme solutions were mixed and incubated at a desired temperature for 10 min. Reactions were then terminated by the addition of 95% ethanol (15 mL), and the released fatty acids were titrated with 0.05 M NaOH. Blanks were measured with a heat-inactivated enzyme sample. The hydrolysis activity at 60 °C of all enzymes in this study was showed in Appendix A.

### 4.5. Regiospecificity Analysis

The determination of the regiospecificity of MAS1 and mutants was performed according to a previously described method [12]. In brief, the hydrolysis of TAGs (including tricaprylin, trilaurin and triolein) by MAS1 and mutants was conducted. All reactions were carried out with continuous shaking at 300 rpm and incubation at a certain temperature. The reaction mixture contained 0.2 g substrate and 10 U purified enzyme dissolved in 1 mL sodium phosphate buffer (0.1 M, pH 7.0). Samples (40 μL) were withdrawn after a 10 min reaction and mixed with 0.5 g of anhydrous sodium sulfate and 1 mL of *n*-hexane, 2-propanol and methanoic acid (13:1:0.003, by volume). Subsequently, the mixture was centrifuged at 10,000× *g* for 1 min, and the supernatant was analyzed using high-performance liquid chromatography to detect the contents of the reactants. The positional specificity index (PSI) was used to evaluate the regiospecificity of MAS1 and its mutants [5]. The formula is as follows:
$$\mathrm{PSI}=\frac{1,2\left(2,3\right)-\mathrm{DAG}-2\;\times \;1,3-\mathrm{DAG}}{1,2\left(2,3\right)-\mathrm{DAG}+2\;\times \;1,3-\mathrm{DAG}}\;\times \;100$$

### 4.6. Model Construction of Enzyme-Substrate Complexes

Since the crystal structure of the lipase from *Streptomyces* sp. W007 (MAS1) was in closed conformation (PDB ID: 5H6B and 5H6G) [12], the lid models of lipases from *Pseudomonas aeruginosa* (PAL, PDB ID: 1EX9) and *Burkholderia cepacia* (BCL, PDB ID: 5LIP), and lipase B from *Candida antarctica* (CaLB, PDB ID: 1LBS) [13,14,22] were employed to build a MAS1 model in an open conformation (Appendix A) by the homology modelling using the MODELLER package (9.22, Andrej Sali, San Francisco, USA) [23]. In the structure of PAL, a TC analogue (octyl-phosphinic acid 1,2-bis-octylcarbamoyloxy-ethyl ester) covalently bound to the catalytic site in the *sn*-1 reaction mode. Thus, the PAL-TC complex model in the *sn*-1 reaction mode was obtained by manually modifying the analogue. The complex model was further minimized in vacuum by using Gromacs 5.14 package (GROMACS development team) [24]. Meanwhile, the force field parameters of the TC molecule was generated from the AMBER GAFF force field [25], and their partial atomic charges were obtained from the restrained electrostatic potential (RESP) charge at the HF/6-31G (d) level with the Gaussian 09 package (E.01, Gaussian, Inc., Wallingford, CT, USA) [26]. Again, the MAS1 in an open conformation and the minimized PAL-TC complex model was used as template to construct the MAS1-TC complex model in the *sn*-1 reaction mode by the homology modelling. The model was minimized in vacuum to decrease the system energy.

To analyze the conformational discrimination among different reaction modes of TAGs, TC in the *sn*-3 reaction conformation was also modelled by a substituent exchange at the C2 position of the glycerol moiety (Appendix A). TC in the *sn*-2 reaction conformation bound to MAS1 was constructed by retaining the *sn*-1 moiety of TC in the *sn*-1 reaction mode as the *sn*-2 moiety of TC in the *sn*-2 reaction mode. Then, the *sn*-1 and *sn*-2 moieties were supplemented manually (Appendix A). Finally, the MAS1-TC complexes in all reaction modes were minimized in vacuum to eliminate the steric clash and normalize the bond parameters of the substrate.

The open conformations of G40E, G40F, G40Q, G40R, G40W, G40Y, N45Y, H108F, H108W, T237R, T237Y and complex models of TC in the *sn*-2 reaction conformation bound to mutants N45A, N45D, N45E, N4R, H108A, H108E, H108Q, T237A, T237E, T237F and T237Q were built by homology model using the corresponding models of wild type built before.

### 4.7. Preparation of Simulation Systems

With the development of biocomputational techniques, more details and key information in regard to the structure–function relationship of lipases have been revealed through simulating the processes of substrate recognition and reaction [27,28]. Before the molecular dynamics simulations, the protonation states of the charged residues in the above built models were firstly determined by the PROPKA 3.1 program (Jensen Group, Copenhagen, Denmark) and their individual local hydrogen-bonding networks were checked manually [29]. The residues H36/H139/H232 and H28/H77/H108/H242 were determined as single protonated on the δ site and ε site, respectively. After the protonation determination, each prepared model was solvated in a cubic water box with a 12 Å buffer distance between the solvent box wall and the nearest solute atoms. Finally, all models were neutralized by the addition of Na^+^ or Cl^−^ at protein surface.

### 4.8. Molecular Dynamic Simulations

All the prepared models were first minimized to relax the solvent and optimize the system. After several steps of minimization, each model was heated to 333 K gradually under the NVT ensemble for 100 ps, followed by another 100 ps of MD simulation under the NPT ensemble (at the temperature of 333 K and the pressure of 1 atm) to relax the system density to about 1.0 g/cm^3^. Afterwards, 50 ns of NPT MD simulation with a time step of 2.0 fs under periodic boundary conditions were performed for each model to produce trajectories via Gromacs 5.14. During the MD process, the TIP3P model and Amber99SB force field were employed for the water molecules and proteins, respectively [30,31,32,33]. The LINCS algorithm was applied to constrain all bonds. The Velocite-rescale and Parrinello–Rahman methods were used to control the system temperature and pressure. A cutoff of 14 Å was set for both van der Waals and electrostatic interactions.

### 4.9. Statistical Analysis

All experiments were carried out in triplicate. The results are reported as the means ± standard deviations (SD).

## Figures and Tables

**Figure 1 ijms-23-05822-f001:**
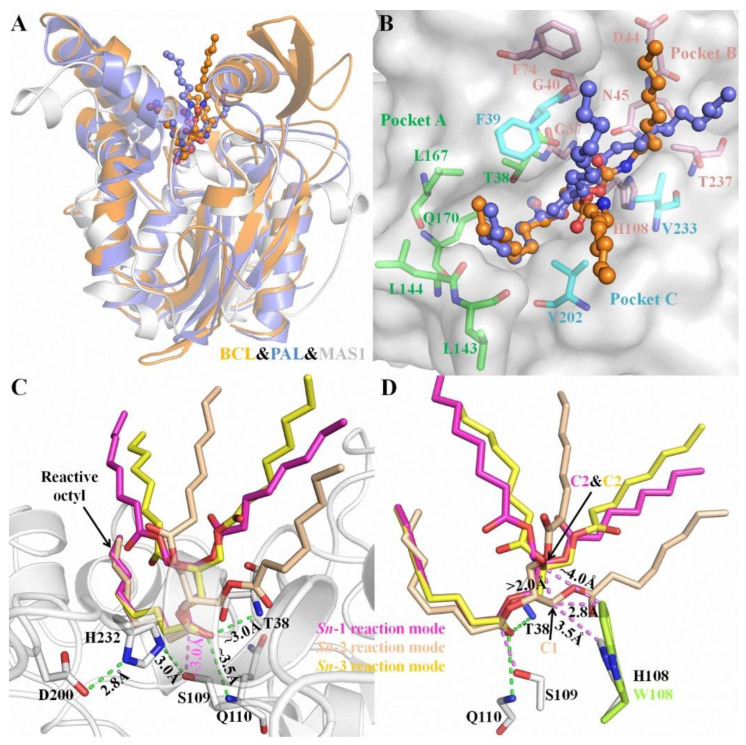
The characteristics of substrate binding with MAS1 in different reaction modes. (**A**) The structural superimposition of BCL (orange), PAL (blue) and MAS1 (white). (**B**) The catalytic pockets of MAS1. The TC analogue from BCL and PAL were showed as orange and blue ball-and-stick, respectively. The comprising residues of pocket A, B and C were showed as green, pink and cyan sticks, respectively. (**C**) The binding conformation of TC in different reaction modes. (**D**) The structural alignment of MAS1 (white) and mutant H108W (lemon). H-bond distances and distances between two atoms were presented as green and pink dashes, respectively.

**Figure 2 ijms-23-05822-f002:**
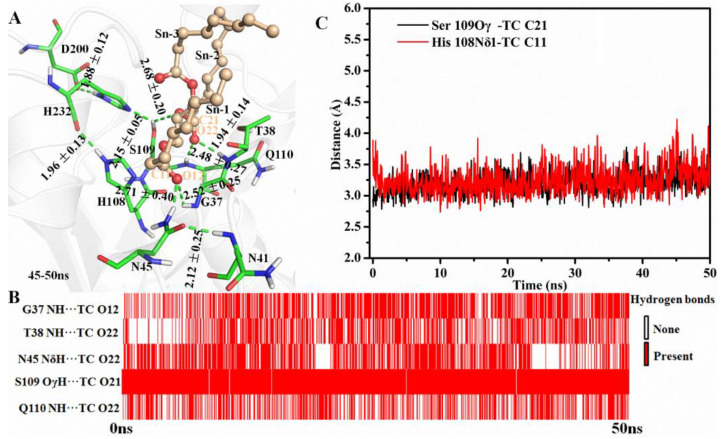
The simulation results of MAS1-TC complex. (**A**) The binding geometry of MAS1 with TC. The H-bonds were presented as green dashes. (**B**) The existence analysis of H-bonds at catalytic site. (**C**) The distance fluctuation analysis of Ser109 Oγ-TC C21 and H108 Nδ1-TC C11.

**Figure 3 ijms-23-05822-f003:**
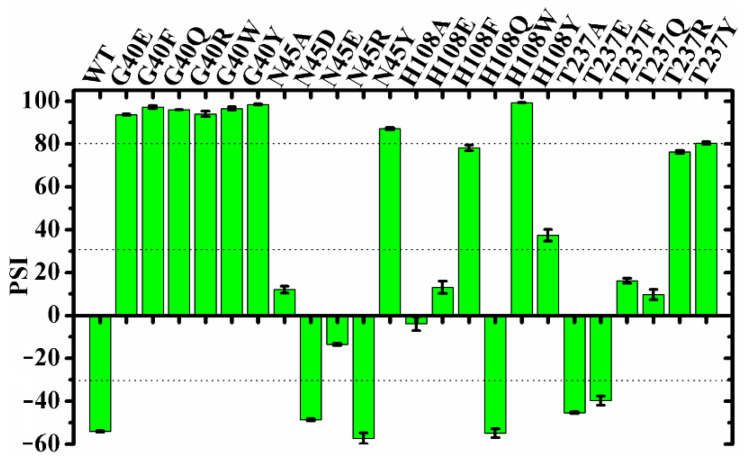
The PSI values of MAS1 and its mutants.

**Figure 4 ijms-23-05822-f004:**
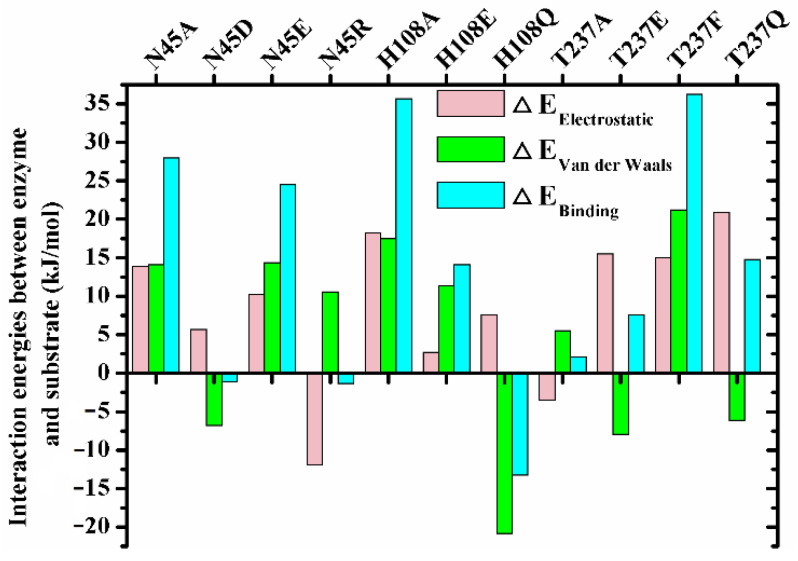
The interaction energies between enzymes and TC during the simulations. $\Delta E_{X}$ = $E_{X}^{\mathrm{Mutant}}-E_{X}^{\mathrm{Wild}-\mathrm{type}}$, where E_X_ represents electrostatic, van der Waals or binding energy obtained from the balanced phase of corresponding simulations (E_binding_ = E_electrostatic_ + E_van der Waals_).

**Figure 5 ijms-23-05822-f005:**
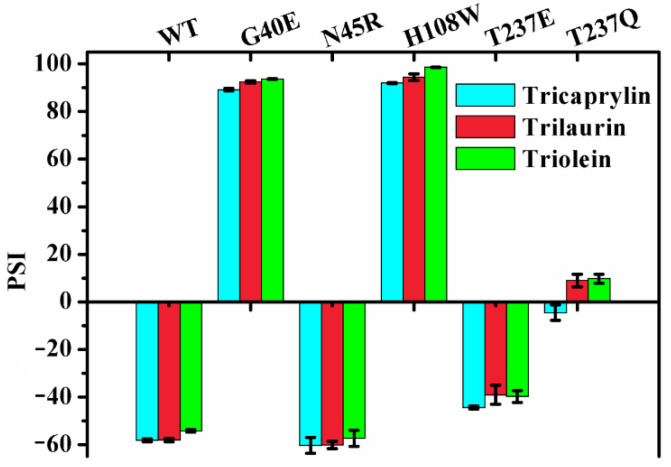
The effect of chain length on regiospecificity of MAS1 and its mutants.

**Figure 6 ijms-23-05822-f006:**
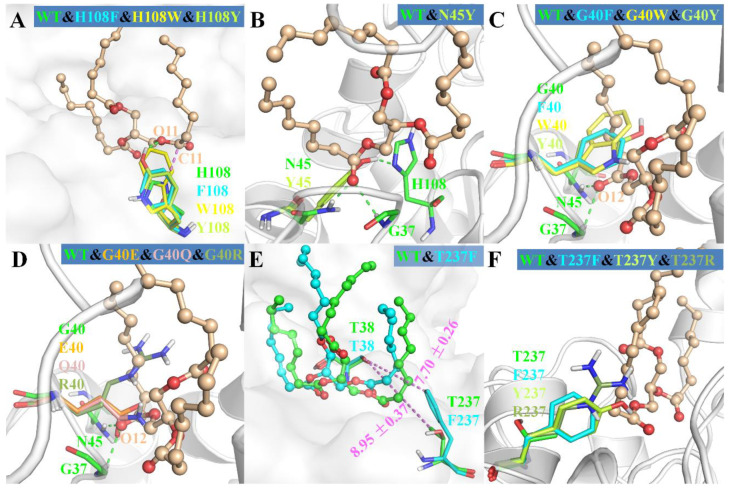
The structural superimposition of mutants with wild-type MAS1. The wild type was showed as cartoon or surface in each figure. The ligand from wild-type complex in wheat ball-and-stick model was showed as the reference in (**A**–**D**,**F**). In (**E**), the ligands from wild type and mutant T237F were colored in green and cyan, respectively.

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
