# Peer review of "Structural Basis for the Regiospecificity of a Lipase from Streptomyces sp. W007"

_ijms, 2022, doi:10.3390/ijms23105822_

Round 1

Reviewer 1 Report

This paper examined the structural basis of the regiospecificity toward TAG of a lipase from Streptomyces sp. W007 (MAS1). Mutagenesis experiments and molecular modelling revealed that the steric hindrance of the site for binding a non-reactive carbonyl group of TAGs could transform the non-specific MAS1 to a α-specific lipase, and altering the local polarity of the site as well as the conformational stability of its composing residues could also impact the regiospecificity. This work is well designed, clearly described, and rationally interpreted, and would be helpful for the rational design of the regiospecificity of lipases. Therefore, this paper is considered acceptable for publishing in its current state except typographical errors listed below:

1. p. 4, line 20: “Figure 2C” seems to be mistaken for “Figure 2B”.

2. p. 7, lines 11-12: “Figure 2A” seems to be mistaken for “Figure 2C”.

3. p. 8, line 4: “Figure 2A” seems to be mistaken for “Figure 2C”.

4. p. 9, line 39: It seems that “the mutant T237Q lost more β-preference than the mutant T237E”.

Author Response

Dear Reviewer,

    We appreciate the constructive comments of you on our manuscript, and the manuscript has been carefully revised according to your comments.

1.The reviewer’s comment: p. 4, line 20: “Figure 2C” seems to be mistaken for “Figure 2B”.

ResposeFigure 2B showed the H-bond interactions between MAS1 and substrate, while Figure 2C showed the distances between atoms in the complex. In p. 4, line 20, we descript the H-bond between Ser109 and TC (Ser109 OγH…TC O21), so Figure 2B was referred in here.

2.The reviewer’s comment: p. 7, lines 11-12: “Figure 2A” seems to be mistaken for “Figure 2C”.

ResposeThanks for your reminder, we have corrected it.

3.The reviewer’s comment: p. 8, line 4: “Figure 2A” seems to be mistaken for “Figure 2C”.

ResposeThanks for your reminder, we have corrected it.

4.The reviewer’s comment: p. 9, line 39: It seems that “the mutant T237Q lost more β-preference than the mutant T237E”.

ResposeThanks for your reminder, we have corrected it.

Reviewer 2 Report

Excellent work. I have no objections or significant remarks, except for minor ones. There are some misprints, for example, in the Discussion section Role of N45, in the 4th line from the top, the last mutant listed is probably N45Y not N45R.

A minor question: what were the reasons for choosing the mutants indicated in Fig. 5 (Section Effect of chain length on the regiospecificity)? In my opinion, it would be nice to explain this in the text.

Author Response

Dear Reviewer,

    We appreciate the constructive comments of you on our manuscript, and the manuscript has been carefully revised.

1.The reviewer’s comment: There are some misprints, for example, in the Discussion section Role of N45, in the 4th line from the top, the last mutant listed is probably N45Y not N45R.

ResposeThanks for your reminder, we have corrected it.

2.The reviewer’s comment: what were the reasons for choosing the mutants indicated in Fig. 5 (Section Effect of chain length on the regiospecificity)? In my opinion, it would be nice to explain this in the text.

ResposeIn here, we wanted to investigate the effect of chain length of fatty acids on regiospecificity of lipases. Thus, lipases with various regiospecificity were selected to examinate (β-selective: WT, N45R and T237E; non-specific: T237Q; α-selective: G40E and H108W). The related content has been added to manuscript as followed:

Previous version (p. 6, line 6-8): Herein, we investigated the effects of chain lengths of fatty acids on the regiospecificity of MAS1 and its mutants including G40E, N45R, H108W, T237E and T237Q with TC, trilaurin and triolein as substrates.

Current version (p. 6, line 8-12): MAS1 and its mutants G40E, N45R, H108W, T237E and T237Q displayed various regiospecificity during catalyzing the hydrolysis of TC (Figure 3). To investigate the effects of chain lengths of fatty acids on lipase regiospecificity, we measured their PSIs using trilaurin and triolein as substrates and compared them with the PSIs determined by hydrolyzing TC.